# Transcriptome Analysis Reveals Inhibitory Effects of Lentogenic Newcastle Disease Virus on Cell Survival and Immune Function in Spleen of Commercial Layer Chicks

**DOI:** 10.3390/genes11091003

**Published:** 2020-08-26

**Authors:** Jibin Zhang, Michael G. Kaiser, Rodrigo A. Gallardo, Terra R. Kelly, Jack C. M. Dekkers, Huaijun Zhou, Susan J. Lamont

**Affiliations:** 1Department of Animal Science, Iowa State University, Ames, IA 50011, USA; jibinzhang12@gmail.com (J.Z.); mgkaiser@iastate.edu (M.G.K.); jdekkers@iastate.edu (J.C.M.D.); 2Population Health and Reproduction, School of Veterinary Medicine, University of California, Davis, CA 95616, USA; ragallardo@ucdavis.edu; 3One Health Institute, School of Veterinary Medicine, University of California, Davis, CA 95616, USA; trkelly@ucdavis.edu; 4Department of Animal Science, University of California, Davis, CA 95616, USA; hzhou@ucdavis.edu

**Keywords:** chicken, Newcastle disease, spleen, immune response, gene expression, RNA-seq

## Abstract

As a major infectious disease in chickens, Newcastle disease virus (NDV) causes considerable economic losses in the poultry industry, especially in developing countries where there is limited access to effective vaccination. Therefore, enhancing resistance to the virus in commercial chickens through breeding is a promising way to promote poultry production. In this study, we investigated gene expression changes at 2 and 6 days post inoculation (dpi) at day 21 with a lentogenic NDV in a commercial egg-laying chicken hybrid using RNA sequencing analysis. By comparing NDV-challenged and non-challenged groups, 526 differentially expressed genes (DEGs) (false discovery rate (FDR) < 0.05) were identified at 2 dpi, and only 36 at 6 dpi. For the DEGs at 2 dpi, Ingenuity Pathway Analysis predicted inhibition of multiple signaling pathways in response to NDV that regulate immune cell development and activity, neurogenesis, and angiogenesis. Up-regulation of interferon induced protein with tetratricopeptide repeats 5 (*IFIT5*) in response to NDV was consistent between the current and most previous studies. Sprouty RTK signaling antagonist 1 (*SPRY1*), a DEG in the current study, is in a significant quantitative trait locus associated with virus load at 6 dpi in the same population. These identified pathways and DEGs provide potential targets to further study breeding strategy to enhance NDV resistance in chickens.

## 1. Introduction

As one of the most devastating avian diseases with worldwide distribution, Newcastle disease has caused significant economic losses in the global poultry industry. As a negative-sense and single-stranded RNA virus, natural evolution of the Newcastle disease virus (NDV) has resulted in a great diversity of strains with varied virulence, from lentogenic and mesogenic to velogenic [1]. Certain velogenic strains can cause up to 100% mortality in a chicken flock, and sometimes even vaccinated chickens are not adequately protected. In many low-income countries around the world, backyard chicken flocks are not provided adequate vaccination and biosecurity [2]. In these countries, chickens play a vital role in supporting people’s livelihoods as an important protein and nutrition source, and poultry raising also plays a critical role in women’s empowerment, with women primarily responsible for decision-making in regards to household chicken production [3]. In addition, NDV outbreaks occur in commercial poultry operations in these NDV endemic regions as a result of improper vaccination programs and challenges associated with maintaining a vaccine cold chain required to administer potent and effective vaccines to flocks. Therefore, improving host resistance to NDV through genetic selection and breeding is a promising complementary strategy to improve poultry production and livelihoods.

The use of genetic selection to improve poultry health is feasible because of the apparent genetic control of the immune response to NDV, as manifested in different responses to NDV across different chicken lines [4,5,6]. Previously, we have shown moderate to high heritability of NDV response traits, including viral load at 2 and 6 days post inoculation (dpi) and antibody titer pre-challenge and at 10 dpi in a commercial layer chicken hybrid [7]. In addition, through a genome-wide association study (GWAS), we have identified major single nucleotide polymorphisms (SNPs) with over 20% genome-wide significance for each of these traits after Bonferroni correction [7]. However, it is still unclear if and how these SNPs affect these traits by regulating gene expression. Previous gene expression studies with inbred chicken lines that have distinct responses to NDV [4,8,9,10,11,12] have contributed to understanding gene regulation of Newcastle disease resistance, but these studies did not reflect commercial settings in two aspects. First, each chicken line had a quite uniform genetic makeup, which rarely exists in commercial production. Second, the parents of these chicks did not receive NDV vaccination, so there were no maternal antibodies against NDV in these chickens. Therefore, we investigated gene expression through RNA sequencing analysis in commercial brown layer chickens. Our objectives were to generate a clearer picture of the transcriptomic response to NDV in commercial chickens and to identify critical genes that regulate resistance to NDV by combining the current RNA-seq and prior GWAS results. The important genes and pathways identified in this study represent a further step toward a selective breeding program for enhancing Newcastle disease resistance in chickens.

## 2. Materials and Methods

### 2.1. Animals and Experimental Design

This study was approved by the Iowa State University Institutional Animal Care and Use Committee (IACUC log number 1-13-7490-G). All experiments were performed in accordance with the relevant guidelines and regulations. The animal experiment was described previously [7]. Three hatches of 200 mixed-sex Hy-Line Brown chicks (*N* = 600) were produced by 16 sires and 145 dams. They were separated based on pedigree information to distribute half-sibs into three different ABSL2 rooms. After being raised with ad libitum access to feed and water until 21 days old, the chickens were either treated with the type B1 LaSota lentogenic strain of NDV (200 μL of 10^8^ EID50 for each chicken, *n* = 540) through nasal and ocular inoculation routes (50 μL into each eye and nostril), or 200 μL phosphate-buffered saline (PBS, *n* = 60) through the same routes. Chickens of different genders were selected for RNA-seq within each treatment group at 2 and 6 dpi (*n* = 4/group/time) and euthanized with sodium pentobarbital. Spleen tissues were collected and placed into RNAlater solution (Thermo Fisher Scientific, Waltham, MA, USA) for short-term storage and transferred to a −80 °C freezer.

### 2.2. Total RNA Isolation

RNA samples were isolated from spleens using Ambion RNAqueous Total RNA Isolation Kit (Thermo Fisher Scientific) following the manufacturer’s protocol and were then treated with DNase using TURBO DNA-*free* kit (Thermo Fisher Scientific). Proper quantity and high quality (RNA integrity number >8) of the RNA samples were ensured through assessment by a NanoDrop ND-1000 UV-vis spectrophotometer (Thermo Fisher Scientific) and based on the RNA 6000 Nano kit in the Agilent 2100 Bioanalyzer (Agilent Technology, Santa Clara, CA, USA), respectively.

### 2.3. cDNA Library Construction and Sequencing

The transcriptome library of each sample was constructed with 0.5 µg total RNA using the Illumina TruSeq RNA sample preparation kit (Illumina Inc., San Diego, CA, USA) following the low sample protocol in the TruSeq RNA sample preparation v2 guide (Part #15026495, March 2014). The cDNA in each library was assayed for proper length by running the DNA 1000 assay on the Agilent 2100 Bioanalyzer. Libraries from 2 and 6 dpi were independently constructed and sequenced at different times. Two libraries randomly selected from each treatment group were multiplexed and pooled into one lane for sequencing, and the other 2 libraries were similarly placed in the other lane, yielding 2 lanes for each time point. Sequences of 100-bp single-end reads were obtained using the HiSeq 2500 Sequencing System (Illumina) at the Iowa State University DNA facility.

### 2.4. Sequence Reads Quality Control, Mapping, and Counting

After quality assessment of raw reads using FastQC (version 0.10.1, https://www.bioinformatics.babraham.ac.uk/projects/fastqc/), adapter sequences and sequences of low quality (Sanger base quality < 20) were trimmed using the FASTX-Toolkit in CyVerse (https://de.cyverse.org/de). The filtered reads of each sample were then aligned to the Gallus gallus Galgal 6.0 reference genome (assembly GCA_000002315.5) from Ensembl using TopHat2 [13] (version 2.1.1) with default parameters. The mapped reads were subsequently counted using HTSeq [14] (version 0.6.1), using default parameters and the *Gallus gallus* Galgal6.0 GTF file from Ensembl. For the unmapped reads, we used them to align with the NDV LaSota sequence (GenBank accession number JF950510.1) with BWA [15] (version 0.7.15) and counted the reads from virus with HTSeq following the procedures described previously [4].

### 2.5. Differential Expression, Pathway, and Co-Expression Analyses

Because the library construction and sequencing steps were done separately for each time point, the differential expression analyses for the NDV-challenged vs. non-challenged contrast were performed separately at 2 dpi and 6 dpi using the edgeR package [16] (version 3.12.1) in R (version 3.2.3). The trimmed mean of M-value (TMM) method was used to normalize read counts. A generalized linear model (GLM) based on a negative binomial distribution was fitted with the main effect of treatment. The Benjamini–Hochberg method was used to control the false discovery rate (FDR). Principal component analysis (PCA) plots were generated for each time point by the ggbiplot program in R. Differentially expressed genes (DEGs) with FDR <0.05 were included in the Ingenuity Pathway Analysis software (IPA, Qiagen, Redwood City, CA, USA), and pathways or functions with |*z*-score| >2 were considered to be activated or inhibited.

### 2.6. Fluidigm Biomark Assay of Gene Expression

To validate RNA-seq results, the Biomark Dynamic assay (Fluidigm, South San Francisco, CA, USA) was used to measure gene expression in the same 16 RNA samples as used in the RNA-seq study. Based on their Log2FC in RNA-seq analysis, we selected 40 genes that covered the full range of Log2FC in the comparisons between two treatments. In addition, the functional importance of each gene in immune defense was considered in the selection of genes. Primers for these genes were previously published [8]. The geometric means of Ct values of 3 housekeeping genes [glyceraldehyde-3-phosphate dehydrogenase (*GAPDH*), actin beta (*ACTB*), and hypoxanthine phosphoribosyltransferase 1 (*HPRT1*)] were used for normalization. For each sample, cDNA was prepared from 50 ng of RNA using Reverse Transcription Master Mix and pre-amplified at 12 cycles using Preamp Master Mix (Fluidigm), according to the manufacturer’s protocols. Gene expression detection in each sample was performed in duplicate using the Fluidigm 48.48 integrated fluidic circuit (IFC). Two IFCs were run in the BioMark HD (Fluidigm) Real-Time PCR system for the 2 dpi and 6 dpi samples, and data were analyzed using the Fluidigm Real-Time PCR Analysis software. Replicates of the same sample that showed a shifted peak in melting curves were removed from further analysis. Gene expression was compared between treatments using the 2(−ΔΔCt) method in Excel, and the pairwise correlation and linear regression between the Log_2_FC results from the Biomark assay and from RNA-seq were calculated using the JMP Pro 12.0.1 software (SAS institute, Cary, NC, USA).

### 2.7. Data Availability Statement

The raw sequencing data generated and analyzed in the current study are available in ArrayExpress repository in EMBL-EBI (http://www.ebi.ac.uk/arrayexpress/experiments/E-MTAB-8350/) with accession number E-MTAB-8350.

## 3. Results

### 3.1. Gene Expression Changes Induced by NDV

Millions of 100-bp single-end reads were obtained for each sample from sequencing of libraries constructed with RNA samples from spleen. Among them, only 1–2 reads were mapped to the virus genome in three NDV-challenged samples, and no reads were mapped for the other samples. Although there was large variation in the number of raw reads among samples after filtering (from 10,383,866 to 16,783,037), similar percentages of reads of each sample were mapped to the Galgal 6.0 reference genome in the Ensembl database (around 90%). On average, expression of 17,255 genes was detected in each sample, accounting for 71% of the 24,356 annotated genes in the reference genome (Appendix A). The Log_2_ fold change (Log_2_FC) of 40 selected genes in different treatment comparisons showed very high consistency between expression data obtained by Fluidigm Biomark qPCR and RNA-seq (correlation coefficient: r = 0.91), validating the differential expression identified by RNA-seq (Figure 1).

The PCA plot shows that samples at 2 dpi separated well between the NDV-challenged and non-challenged groups along principal component 1 (PC1), which explained 28.9% of the variation in RNA-seq data among the samples (Figure 2A). Samples at 6 dpi clustered the best along PC2 and PC3, but there is still one NDV-challenged sample intermingled with non-challenged group (Figure 2B).

The PCA results also agree with the difference in numbers of DEGs for the NDV-challenged vs. non-challenged group between 2 dpi and 6 dpi. At FDR <0.05, expression of 526 genes was significantly changed by NDV treatment at 2 dpi when compared to the non-challenged group. Among these DEGs, 84% of them were downregulated by NDV. At 6 dpi, NDV treatment only induced differential expression of 36 genes (Figure 3A, Appendix A). Five DEGs were shared between the two time points (Figure 2B), which were all downregulated by NDV challenge (Figure 3B). These genes are integrin alpha-8 (*ITGA8*), follicle stimulating hormone receptor (*FSHR*), thrombospondin 1 (*THBS1*), transmembrane protein 47 (*TMEM47*), Rho GTPase activating protein 20 (*ARHGAP20*) (Table 1). However, the fold change of these genes in response to NDV challenge was not large. Instead, Apolipoprotein L Domain Containing 1 (APOLD1), chemokine CCLI7, and Nuclear Receptor Subfamily 4 Group A Member 1 (*NR4A1*) were the top down-regulated genes based on fold change (LFC < −3), while neurturin (*NRTN*) was the top up-regulated gene (LFC > 3) at 2 dpi.

### 3.2. Ingenuity Pathway Analysis of Differentially Expressed Genes

IPA showed significant inhibition (*p* < 0.05, *z*-score < −2) of 52 pathways following NDV challenge at 2 dpi, of which the top pathways (*z*-score < −3) are listed in Table 2. Among these pathways, IL-15 production, Tec Kinase signaling, IL-8 signaling are involved in immune response, while the other pathways are mainly related to cell adhesion, signaling and movement, and neurogenesis and neuropathogenesis (Table 2). Down-regulated genes in response to NDV were involved in several pathways, such as phosphoinositide-3-kinase regulatory subunit 6 (*PIK3R6*), SRC proto-oncogene, non-receptor tyrosine kinase (*SRC*), FOS proto-oncogene, AP-1 transcription factor subunit (*FOS*), Integrin Subunit Alpha 3 (*ITGA3*) and hemopoietic cell kinase (*HCK*) are involved in several pathways.

At 6 dpi, no significant pathway was predicted with |*z*-score| > 2 based on change of gene expression, but some pathways were significantly enriched (*p* < 0.05) by DEGs (Table 3). Among these pathways, Granulocyte adhesion and diapedesis, Agranulocyte adhesion and diapedesis, Chondrotin sulfate degradation, and G-Protein coupled receptor signaling pathways regulate activation of inflammatory response. The DEGs in these pathways were Histamine receptor H1 (*HRH1*), KIT ligand (*KITLG*), Hyaluronidase 2 (*HYAL2*), C-C motif chemokine ligand 21 (*CCL21*), and *THBS1* (Table 2). These genes all regulate phagocyte activation, as shown in Table 3.

Disease and biofunction prediction of IPA suggested decreased cell mobility and viability, decreased immune cell activity, decreased neurogenesis, and angiogenesis in response to NDV at 2 dpi, which together contribute to increased morbidity and mortality of organism at 2 dpi. By 6 dpi, the NDV challenge seemed to still have mild inhibitory effects on immune function and the nervous system, which promotes organismal death (Table 3).

### 3.3. Comparison with Previous RNA-Seq and GWAS Studies

We found 87 genes that were shared between this study and previous reported gene expression changes in response to NDV. Among these genes, 33 (38%) showed consistent expression changes in response to NDV in spleen of Hy-Line Brown birds and in spleen [8,17,18], Harderian gland [9,12], trachea [4], lung [10] or embryo [11] of Fayoumi or Leghorn chickens (Table 4). Notably, interferon induced protein with tetratricopeptide repeats 5 (*IFIT5*) showed significant up-regulation in response to NDV in the spleen of all three chicken lines.

Compared to the previous GWAS study in the population from which the chickens in this study were selected, the most significant overlap is the Sprouty RTK signaling antagonist 1 (*SPRY1*) gene in chromosome 4 (Table 5), which was significantly down-regulated by NDV at 2 dpi and is located within 1Mb downstream of a SNP that was significantly associated with viral load at 6 dpi [7]. In addition, the DEGs filamin binding LIM protein 1 (*FBLIM1*), zinc finger and BTB domain containing 17 (*ZBTB17*), EPH receptor A2 (*EPHA2*) on chromosome 21 are located within 500 kb downstream of a SNP that was significantly associated with antibody titer at 10 dpi [7].

Pannexin 3 (*PANX3*), serine and arginine rich splicing factor 5 (*SRSF5*) and ST3 beta-galactoside alpha-2,3-sialyltransferase 4 (*ST3GAL4*) are another three DEGs located within 1 Mb of SNPs significantly associated with viral load at 2 or 6 dpi in Hy-Line Brown under heat stress [19] or Tanzanian local chickens [20]. In addition, carboxypeptidase N subunit 2 (*CPN2*) is another DEG in this study that is located within 1 Mb of a SNP associated with antibody titer at 10 dpi in Tanzanian local chicken ecotypes [20]. Expression of all these genes is inhibited at 2 dpi by NDV challenge (Table 5).

Comparison of the pathways identified in this and other gene expression studies that investigated response to NDV showed 17 shared significant pathways (Table 6). Interestingly, all these pathways are predicted by IPA to be inhibited (*z*-score < −2) in spleen of Hy-Line Brown chickens but activated in lung [10], trachea [4] and Harderian gland [12] of Fayoumi and Leghorn chickens, although all chickens were treated by the same lentogenic strain of NDV.

## 4. Discussion

NDV is a contagious viral disease that involves complex host-virus interactions in the immune, nervous, and respiratory systems. Chickens infected with NDV show respiratory signs such as gasping and coughing; nervous signs such as tremors, paralyzed wings and legs, and twisted neck; swelling in the eyes and neck [21]; watery diarrhea [22]; and reduced egg and meat production [23]. ND can cause damage to hematopoietic [24], lymphoid [25], vascular and neural cells [26]. NDV can infect and lead to apoptosis of lymphocytes and macrophages, and deactivation of heterophils [27]. Although the virus used in this study is a mildly virulent strain, and there are few viral gene sequences in the RNA sequences, the spleen may have fast and sensitive response to the infection even when the pathogens are at distant sites. Spleen is a crucial hub for bidirectional communication between the nervous and immune system [28]. It contributes to the greatest number of effector T cells to the lung during respiratory infectious disease and the immune response is more correlated with number and kinetics antigen presenting cells rather than the virus load [29]. Therefore, the spleen showed gene expression to the disease possibly through migration of immune cells and systematic activation by neuro system.

Among the top 10 pathways (*z*-score < −3) altered by NDV at 2 dpi, the decreased expression of PIK3R6 contributes to inhibition of eight pathways, because phosphoinositide 3-kinases (PI3K) has an extensive role in cell survival, migration, growth and differentiation [30]. PI3Kγ, which contains PIK3R6, is a kinase in leukocytes play an important role in survival and migration of immune cells and their function and proliferation [31]. Inhibition of PI3K activation has been proven to promote cell apoptosis during early stage NDV infection [32]. Therefore, the downregulation of 3-phosphoinositide biosynthesis pathways may suppress immune cell function and development.

The downregulation of some immune pathways also suggests that the NDV may have caused an immunosuppressive effect. IL-15 production mediates development, homeostasis and anti-viral response of natural killer cells and CD8 T cells [33]. Tec kinase signaling pathway is also required for differentiation and development of CD4+ [34] and CD8+ T cell [35], and CD8+ T cell activation in response to viral infection [36]. IL-8 signaling mainly regulates chemotaxis and activation of neutrophils during infectious disease [37]. Therefore, inhibition of these pathways could lead to a reduced immune response.

The pathway analysis also pointed out some pathways related to the repair of damage to vascular cells and neural cells which is common in ND pathology. Adrenomedullin signaling is a significant pathway regulating endothelial cell apoptosis, angiogenesis and vascular permeability [38]. GP6 signaling is an important pathway that protects vascular integrity and prevents bleeding by recruiting platelets to neutrophil-induced vascular breach during inflammatory response [39]. Inhibition of these pathways may compromise the functional integrity of blood vessels. The gonadotropin-releasing hormone (*GNRH*) is mainly secreted by central nervous system, and GNRH signaling is vital for both central and peripheral reproductive regulation [40]. Synaptogenesis signaling pathway regulates formation of synapses between neurons in nervous system [41]. Reelin signaling in neurons is also very important for neuronal migration and aggregation as well as synapse function [42]. Therefore, the inhibition of these three pathways may be related to nervous damage caused by NDV.

Most genes among the 526 DEGs at 2 dpi showed less than 2 times decrease in their expression, indicating that these predicted negative responses are moderate even if they are statistically significant. There was no mortality in the chickens in this study, and the response was alleviated greatly by 6 dpi. The consistent down-regulation of Rho GTPase activating gene *ARHGAP20*, integrin gene *ITGA8*, *FSHR* and *THBS1* (Table 1) between 2 and 6 dpi suggests constant inhibition of microtubule dynamics (Table 3).

The inhibition of pathways that regulate cellular actin and microtubule dynamics, such as signaling by Rho family GTPases and integrin signaling, may block NDV entry into host cells through fusion of viral envelope with the plasma membrane of target cells or fusion of infected cells with adjacent cell membrane [43]. It has been reported that inactivation of RhoA signaling could inhibit the cell-cell fusion and syncytium formation induced by glycoprotein of respiratory syncytial virus which is a paramyxovirus [44]. Therefore, as a paramyxovirus, NDV may have its replication and infection compromised by down-regulation of Rho family GTPases signaling. However, the inhibition of Rho family GTPases signaling may also impair the lymphocyte development, activation and migration [45], thus not beneficial for host immune defense.

The small proportion of shared DEGs between the current study and previous studies, as well as the opposite regulation of 17 shared significant pathways predicted by IPA suggest quite different response to NDV in different tissues or chicken lines. This is consistent with the previous reports about tissue-specific immune response [46] and distinct immune gene expression between different chicken breeds [47]. Maternally transferred antibody may also be a contributing factor to gene expression differences among studies. Because the Hy-Line Brown chickens in this study are commercial chickens whose dams had received 5 immunizations for NDV before production of the chicks, they had detectable circulating maternal antibody at the time of NDV challenge [7]. However, Fayoumi and Leghorn chickens in other studies [4,9,10,12] did not have detectable maternal antibody at the time of experiment. It has been shown that maternal antibody has a blocking effect on the immune response to NDV vaccination in chickens [48,49] and kittiwakes [50]. Although the trans-generation transfer of antibody is believed to protect the neonates during a vulnerable period and save the energy of costly innate immune response for growth and development, it could suppress humoral immunity by binding and hiding antigen epitope from neonatal B-cells [51]. Therefore, the difference of maternal antibody may also contribute to the differential expression profiles among these studies.

On the other hand, those genes showing consistent expression regulation despite many varied experimental factors may play a universal role in immune response to NDV. One of the most significant of those genes is *IFIT5* (Table 4). As an interferon-stimulated gene, IFIT5 has been shown to be critical for innate immune defense against virus [52] by recognizing and inhibiting translation of viral RNA bearing a 5′-triphosphate [53], and positively regulating of nuclear factor kappa-light-chain-enhancer of activated B-cells (NF-κB) signaling [54]. Transgenic chickens with overexpression of IFIT5 showed significant enhanced resistance to velogenic NDV, with remarkable reduction of chickens with clinical signs when challenged with a clinical dose (10E5 EID50) and a 4-day delay of death when challenged with lethal dose (10E6 EID50) [55].

The finding of *SPRY1* as a DEG induced by NDV and a positional candidate gene associated with NDV viral load in GWAS in the same animal experiment indicates its significance in the regulation of response to NDV. SPRY1, as a modulator of fibroblast growth factor, epidermal growth factor signaling [56] and receptor tyrosine kinase signaling pathways [57], has a ubiquitin role in tissue development and critical role for proper function. It is an important regulator for angiogenesis [58], vascular smooth muscle cell differentiation [59], hematopoiesis [60], T cell proliferation [61], splenic erythropoiesis [62], uretic [63] and mammary [56] epithelial branching morphogenesis. Ablation of *SPRY1* has been shown to increase splenic erythropoiesis [62], as well as survival and activity of CD4+ and CD8+ T cells [61,64]. Therefore, the downregulation of *SPRY1* may be a positive response against the viral infection. *FBLIM1*, *ZBTB17* and *EPHA2* are DEGs in this study and positional candidate genes associated with antibody titer in the same animal experiment. Mutation of *ZBTB17* has been reported to block early intrathymic T cell development [65]. *EPHA2* is an important gene regulating neurogenesis [66], angiogenesis [67], integrin-mediated cell adhesion and cell migration [68]. Therefore, they may have important function in immune defense and tissue repair during NDV challenge. *CPN2*, *SRSF5*, *ST3GAL4*, *PANX3* are DEGs in vicinity of SNPs associated with antibody titer or viral load in other NDV studies under artificial [19] or natural [20] heat stress, suggesting the necessity of scrutiny in their functions in protection of immune system against suppression of heat stress.

## 5. Conclusions

In conclusion, we observed extensive negative regulation of gene expression in response to NDV in the spleen of Hy-Line Brown chickens, although this seems mitigated from 2 dpi to 6 dpi. The quite different gene expression responses to NDV vaccine in this study compared to the other previous studies may be due to the difference of tissues, chicken population, and maternal antibody. Through comparison of DEGs in this study with previously identified DEGs and positional candidate genes from GWAS, we identified some critical genes that regulate response to NDV in chickens. This is a further step toward understanding the molecular regulation of response to NDV challenge and identifying the potential key targets to enhance NDV resistance through breeding in poultry.

## Figures and Tables

**Figure 1 genes-11-01003-f001:**
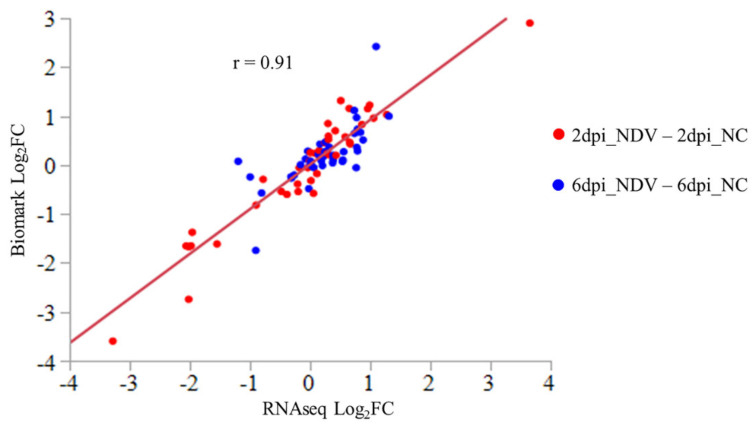
Gene expression analysis by Fluidigm Biomark assay of log_2_ fold change (Log_2_FC) of selected genes that were significant for different contrasts in RNA-seq analysis. Contrasts between treatments at different time points are marked in different colors and each combination is labeled as Time_Treatment (dpi: day post inoculation; NDV: NDV-challenged, NC: Non-challenged). Pearson correlation coefficient is labeled as “r”. Log_2_FC in Biomark assay equals −ΔΔCT for each comparison. Average expression of three housekeeping genes, *GAPDH*, *ACTB*, and *HPRT1,* was used for normalization of Ct values.

**Figure 2 genes-11-01003-f002:**
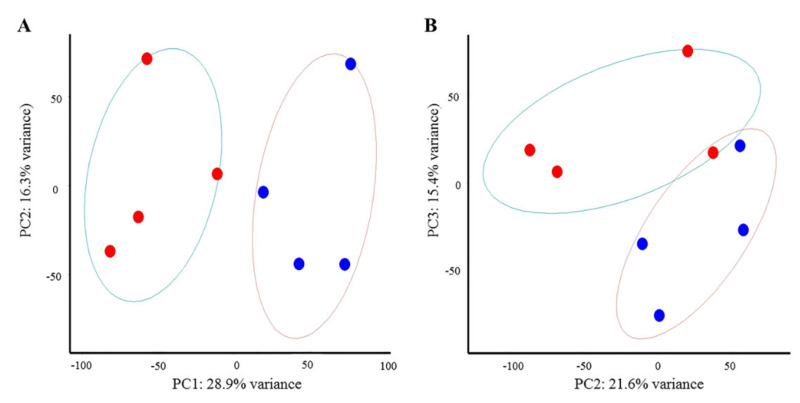
Principal component analysis (PCA) plots generated with ggbiplot in R showing variation and separation of samples between different treatments. (**A**) PCA plot for samples at 2 days post inoculation (dpi). (**B**) PCA plot for samples at 6 dpi. Sample in different treatments are separated by a line. Different treatments are represented in different colors: NDV-challenged (red), non-challenged (blue).

**Figure 3 genes-11-01003-f003:**
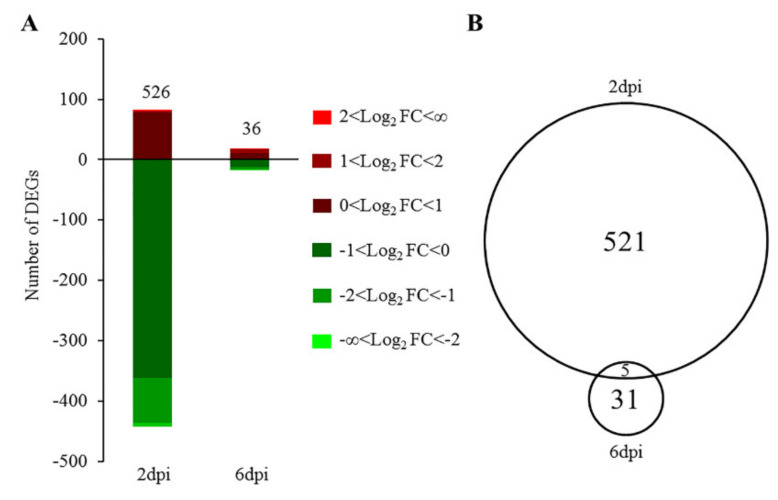
Number of significant differentially expressed genes (DEGs) for NDV-challenged vs non-challenged birds at false discovery rate <0.05. (**A**) Histogram showing the number of DEGs at 2 and 6 days post inoculation (dpi), and their distribution. Up-regulated and down-regulated DEGs are represented in red and blue color. DEGs within different ranges of Log_2_ fold change are represented in different brightness of the color. (**B**) Venn diagram showing the number of overlapping DEGs at 2 and 6 dpi.

**Table 1 genes-11-01003-t001:** Shared DEGs (*p* < 0.05) for NDV-challenged vs. Non-challenged at 2 and 6 days post inoculation.

Ensembl ID	Gene Name	2 Dpi		6 Dpi	
		Log_2_FC	FDR	Log_2_FC	FDR
ENSGALG00000008747	*ITGA8*	−0.36	0.04896	−0.41	0.04412
ENSGALG00000009100	*FSHR*	−0.36	0.04813	−0.45	0.04412
ENSGALG00000009626	*THBS1*	−0.67	0.01366	−0.42	0.03912
ENSGALG00000027198	*TMEM47*	−0.43	0.01800	−0.52	0.00341
ENSGALG00000036566	*ARHGAP20*	−0.41	0.00242	−0.67	0.00002

Log_2_FC represents log2 fold change. FDR means false discovery rate.

**Table 2 genes-11-01003-t002:** Top significant pathways (*p* < 0.05) predicted by IPA with differentially expressed genes (DEGs) for the NDV-challenged vs. Non-challenged comparison.

Time	Pathways	Top DEGs Contributing to Prediction	Ratio	*z*-Score
2 dpi	IL-15 production	*EPH(A2/A5/B1), ROR1, MATK, SRC, DDR2, FES, HCK, AATK*	15/121	−3.357
	Signaling by Rho family GTPases	*FOS, ARHGEF(11/16/17), ACTA1, ITGA3, JUN, LIMK1, PIK3R6*	17/244	−3.153
	GNRH signaling	*HBEGF, FOS, EGR1, MAP3K(6/14), CAMK2B, CACNA2D2, SRC*	14/173	−3.317
	Synaptogenesis signaling pathway	*EPH(A2/A5/B1), GRIN3B, CAMK2B, THBS1, SRC, HCK, PIK3R6*	19/312	−4.000
	Integrin signaling	*ACT(A1/N4), BCAR1, ITGA(3/8), SRC, VCL, MPRIP, RHOB, PIK3R6*	15/213	−3.606
	Reelin signaling in neurons	*GRIN3B, ARHGEF(11/16), ITGA3, CAMK2B, SRC, HCK, PIK3R6*	10/129	−3.162
	GP6 signaling pathway	*LAM(B2/A5), COL(5A1/6A3/24A1), PIK3R6*	9/119	−3.000
	Tec kinase signaling	*FOS, ACTA1, ITGA3, TNFRSF25, SRC, HCK, RHOB, FGR, PIK3R6*	9/164	−3.000
	Adrenomedullin signaling pathway	*FOS, IL1RN, PLCD3, KCNH2, MATK, NPR2, ITPR3, PIK3R6*	10/197	−3.000
	IL-8 signaling	*HBEGF, FOS, SRC, JUN, RHOB, LIMK1, PLD1, FLT4, PIK3R6*	9/200	−3.000
6 dpi	Hematopoiesis from multipotent stem cells	*KITLG*	1/12	-
	Granulocyte adhesion and diapedesis	*HRH1*, ***CCL21***	2/180	-
	Chondrotin sulfate degradation	***HYAL2***	1/16	-
	Agranulocyte adhesion and diapedesis	*HRH1*, ***CCL21***	2/193	-
	Dermatan sulfate degradation	***HYAL2***	1/17	-
	G-Protein coupled receptor signaling	*FSHR, HRH1*	2/272	-
	Inhibition of angiogenesis by TSP1	*THBS1*	1/34	-

Bold italic and italic text respectively indicate higher and lower expression of differentially expressed genes (DEGs) in NDV-challenged chickens compared to non-challenged group. Genes are ranked ascendingly based on their fold change from left to right. Genes within the same family are labeled with the member or subunit names in the brackets. Ratio = (number of DEGs in a pathways)/(Total number of genes in the pathway).

**Table 3 genes-11-01003-t003:** Top disease and functions (*p* < 0.01) predicted by IPA for NDV-challenged vs. Non-challenged comparison.

Time	Disease and Biofunctions	Top DEGs Contributing to Prediction	*z*-Score	No.
2 dpi	Organismal death	*STAR, NR4A3, KLF4, SIK1, HBEGF, CCN1, ADAMTS4, FOS, IL1RN, DUSP1*	9.770	159
	Necrosis	***NRTN*,** *STAR, NR4A3, KLF4, CCN1, IL1RN, DUSP1, ASTN1, EPHA2, ADAM15*	2.841	162
	Viral Infection	*EGR1, CROCC, MAP3K12, FBLIM1, ARTN, DPP4, FAM167B, BCAR1, BMP1*	−5.663	90
	Cell viability	*NR4A(1/3), HBEGF, CCN(1/2), FOS, DUSP1, CACNB3, WNT5A, EGR1, EPHA2*	−5.568	98
	Migration of cells	*CCL24, PDCH15, HBEGF, CCN1, FOS, ITGA2B, WNT5A, FOSL2, EGR1, EPHA2*	−5.554	142
	Vasculogenesis	*CCL24, NR4A1, HBEGF, ATF3, CCN(1/2), FOS, WNT5A, EGR1, EPHA2, ADAM15*	−5.154	66
	Microtubule dynamics	*ATF3, CCN(1/2), FOS, WNT5A, EGR1, EPHA2, CROCC, METRN, GEM, BCAR1*	−4.553	110
	Endocytosis	*HBEGF, WNT5A, EGR1, PEAR1, FLNA, CSF1, RASA4, THBS1, SRC, HIP1, TGM2*	−4.147	43
	Development of neurons	*CCN1, ADAMTS4, FOS, WNT5A, ARTN, METRN, GEM, ITGA3, NR2F1, EGFL7*	−3.640	65
	Leukocyte migration	*CLL24, HBEGF, CCN(1/2), FOS, ITGA2B, WNT5A, FOSL2, EPHA2, ADAM15*	−2.976	69
	Phagocytosis of cells	*WNT5A, PEAR1, FLNA, RASA4, THBS1, SRC, SCARF1, TGM2, LRP1, HCK, LIMK1*	−2.398	26
6 dpi	Proliferation of lymphocytes	*HRH1, KITLG, ZEB2, THBS1*, ***CCL21***	−1.938	5
	Development of neurons	*FSHR, KITLG, ZEB2, THBS1*, ***CCL21***	−1.706	5
	Recruitment of cells	*HRH1, KITLG, THBS1*, ***CCL21***	−1.253	4
	Activation of phagocytes	*HRH1, KITLG, THBS1*, ***CCL21, HYAL2***	−1.188	5
	Microtubule dynamics	*KITLG, ZEB2, FSHR, SPATA13, THBS1, ITGA8*, ***CCL21***	−1.083	7

Bold italic and italic text respectively indicates higher and lower expression of differentially expressed genes (DEGs) in NDV-challenged chickens compared to non-challenged group. Genes are ranked ascendingly based on their fold change from left to right. Genes within the same family are labeled with the member or subunit names in the brackets. No. means number of DEGs related to the disease and biofunction.

**Table 4 genes-11-01003-t004:** DEGs with consistent responses in expression to NDV in the current and previous studies.

Gene Name	Time	LFC	Consistent Response in Other NDV Studies
*IFIT5*	2 dpi	1.27	Spleen of Leghorn at 1 [17], 2 [8,17] and 6 dpi [8], and of Fayoumi at 2 dpi [8]
*CCL19*	6 dpi	0.84	Trachea of Fayoumi and Leghorn at 2 dpi [4]
*AICDA*	2 dpi	0.80	Trachea of Leghorn at 6 dpi [4]
*SLBP*	2 dpi	0.68	Spleen [8] and trachea [4] of Leghorn at 2 dpi
*TRIM24*	2 dpi	0.52	Trachea of Leghorn at 6 dpi [4]
*PLCXD1*	2 dpi	0.50	Harderian gland of Leghorn at 6 dpi [9]
*SNX10*	2 dpi	0.50	Spleen [8] and trachea [4] of Leghorn at 2 dpi
*DRAM1*	2 dpi	0.49	Trachea of Fayoumi and Leghorn at 2 dpi [4]
*FGL2*	2 dpi	0.44	Trachea of Leghorn at 2 dpi [4]
*ASNS*	2 dpi	0.40	Harderian gland of Leghorn at 6 dpi [9]
*PARP12*	2 dpi	0.40	Harderian gland at 2 and 6 dpi [9], in spleen at 2 dpi [8] in Leghorn
*SEPT2*	2 dpi	0.38	Trachea of Fayoumi at 2 and 6 dpi and Leghorn at 6 dpi [4]
*BFAR*	2 dpi	0.37	Trachea of Fayoumi and Leghorn at 2 dpi [4]
*OSTM1*	2 dpi	0.34	Trachea of Leghorn at 2 dpi [4]
*ARHGAP15*	2 dpi	0.33	Trachea of Fayoumi at 2 dpi and Leghorn at 2 and 6 dpi [4]
*CDC42SE2*	2 dpi	0.33	Trachea of Fayoumi at 2 dpi [4]
*P2RY8*	2 dpi	0.33	Harderian gland at 6 dpi [9] and trachea at 2 and 6 dpi [4] in Leghorn
*MYH10*	2 dpi	−0.37	Trachea of Leghorn at 6 dpi [4]
*FSHR*	2 and 6 dpi	−0.40	Trachea of Leghorn at 6 dpi [4]
*SEPT11*	2 dpi	−0.41	Trachea of Leghorn at 6 dpi [4]
*OGN*	6 dpi	−0.44	Trachea of Leghorn at 6 dpi [4]
*KAZALD1*	2 dpi	−0.47	Trachea of Leghorn at 6 dpi [4]
*SRSF5*	2 dpi	−0.50	Trachea of Leghorn at 2 dpi [4]
*P2RX1*	2 dpi	−0.51	Lung of Fayoumi at 10 dpi [10]
*HPSE2*	2 dpi	−0.52	Trachea of Leghorn at 2 dpi [4]
*JUN*	2 dpi	−0.55	Leghorn embryo [11]
*EPHB1*	2 dpi	−0.57	Trachea of Fayoumi at 2 dpi [4]
*LMOD1*	2 dpi	−0.65	Lung of Fayoumi at 10 dpi [10]
*ROR1*	2 dpi	−0.73	Lung of Fayoumi at 2 dpi [10]
*ST3GAL4*	2 dpi	−0.77	Harderian gland of Leghorn at 6 dpi under heat stress [12]
*UROC1*	2 dpi	−1.01	Harderian gland of Leghorn at 6 dpi [9]

**Table 5 genes-11-01003-t005:** DEGs near SNPs identified to be associated with NDV disease traits in previous GWAS studies.

Gene Name	Time	LFC	Trait Associated with Nearby SNPs Identified in Previous Study
*EPHA2*	2 dpi	−1.30	Antibody titer at 10 dpi in GWAS with the same population [7]
*FBLIM1*	2 dpi	−1.09	Antibody titer at 10 dpi in GWAS with the same population [7]
*PANX3*	2 dpi	−0.88	NDV titer at 6 dpi in GWAS with Hy-Line Brown chickens under heat stress [19] and Tanzanian chicken ecotypes [20]
*ST3GAL4*	2 dpi	−0.77	NDV titer at 6 dpi in GWAS with Hy-Line Brown chickens under heat stress [19] and Tanzanian chicken ecotypes [20]
*SPRY1*	2 dpi	−0.69	Viral load at 6 dpi in GWAS with the same population [7]
*CPN2*	2 dpi	−0.65	Antibody to NDV at 10 dpi in GWAS with Tanzanian chicken ecotypes [20]
*SRSF5*	2 dpi	−0.50	NDV titer at 2 dpi in GWAS with Tanzanian chicken ecotypes [20]
*ZBTB17*	2 dpi	−0.33	Antibody titer at 10 dpi in GWAS with the same population [7]

**Table 6 genes-11-01003-t006:** Pathways found to be significantly inhibited by NDV in the present study (*z*-score < −2) and identified to be activated by NDV in other studies, as predicted by IPA.

Pathway	*z*-Score	Activation Predicted by IPA in Other Studies
Integrin signaling	−3.61	Lung of Fayoumi at 10 dpi [10]
IL-8 signaling	−3.00	Lung of Fayoumi at 10 dpi [10] and trachea of Fayoumi and Leghorn at 2 dpi [4]
Tec kinase signaling	−3.00	Lung of Fayoumi at 10 dpi [10] and trachea of Fayoumi and Leghorn at 2 dpi [4]
GP6 signaling pathway	−3.00	Harderian gland of Leghorn at 6 dpi under heat stress [12]
B cell receptor signaling	−2.83	Trachea of Fayoumi and Leghorn at 2 dpi [4]
IL-6 signaling	−2.83	Trachea of Fayoumi and Leghorn at 2 dpi [4]
Production of nitric oxide and reactive oxygen species in macrophage	−2.83	Trachea of Fayoumi and Leghorn at 2 dpi [4]
P2γ purigenic receptor signaling pathway	−2.45	Lung of Fayoumi at 10 dpi [10]
Fcγ receptor-mediated phagocytosis in macrophages and monocytes	−2.45	Trachea of Fayoumi and Leghorn at 2 dpi [4]
Thrombin signaling	−2.33	Lung of Fayoumi at 10 dpi [10]
Ephrin receptor signaling	−2.24	Lung of Fayoumi at 10 dpi [10]
Relaxin signaling	−2.24	Lung of Fayoumi at 10 dpi [10]
CD40 signaling	−2.24	Trachea of Fayoumi and Leghorn at 2 dpi [4]
Leukocyte extravasation signaling	−2.12	Trachea of Fayoumi and Leghorn at 2, 6 and 10 dpi [4]
TNFR1 signaling	−2.00	Trachea of Fayoumi and Leghorn at 2 dpi [4]
TNFR2 signaling	−2.00	Trachea of Fayoumi and Leghorn at 2 dpi [4]

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
