# Peer review of "Transcriptome Analysis Reveals Inhibitory Effects of Lentogenic Newcastle Disease Virus on Cell Survival and Immune Function in Spleen of Commercial Layer Chicks"

_genes, 2020, doi:10.3390/genes11091003_

Round 1

Reviewer 1 Report

Transcriptome analysis reveals inhibitory effects of lentogenic Newcastle disease virus on cell survival and immune function in spleen of commercial layer

Summary

This article focusses on the changes in gene expression of commercial layers, especially analysing those linked to immune function and cell survival, when challenged with lentogenic Newcastle disease virus (NDV). It compares expression changes between 2 and 6 days post infection, and shows a down regulation in a number of immune response linked genes as well as those involved in cell signalling and repair. The data taken from this study is hoped to be used to further future studies looking into breeding strategies to combat or reduce infections linked to NDV.

Broad Comments:

This article is well organised and attempts to answer the current and interesting topic of gene expression linked to a particularly challenging and damaging virus. The gene expression data appears well thought through, and could help to forward our understanding of how the chicken immune system responds to vaccinations and viral challenges. It is however, unfortunate that there was no control group containing non-vaccinated parent animals provided. Especially, as it was stated that in many low-income countries vaccinations are less likely to be adequately provided. Incorporating this into the study would have given it further depth and broadened the appeal of the work. 

Specific comments

  1. Line 250-253: “Chickens infected with NDV show respiratory signs such as gasping and coughing; nervous signs such as tremors, paralyzed wings and legs, and twisted neck; swelling in the eyes and neck; watery diarrhea; and reduced egg and meat production”

This statement is missing references.

  1. Line 265-266: “The immunosuppressive effect of the NDV vaccine has also manifested in downregulation of some immune pathways”

This statement assumes that the immunosuppressive effect is proven, if this is the case it should be previously stated. I would believe that a suggestive statement of “may cause an immunosuppressive effect” would be better suited unless there is also corresponding blood work provided.

  1. 293-294: “Inhibition of Rho family GTPases pathway has been shown to inhibit the cell-cell fusion induced by glycoprotein of respiratory syncytial virus”

Could this statement be expanded on? Perhaps what is inferred by cell-cell fusion, e.g wound healing or damage repair? And in addition, how this could be useful in this model?

Author Response

Broad Comments:

This article is well organised and attempts to answer the current and interesting topic of gene expression linked to a particularly challenging and damaging virus. The gene expression data appears well thought through, and could help to forward our understanding of how the chicken immune system responds to vaccinations and viral challenges. It is however, unfortunate that there was no control group containing non-vaccinated parent animals provided. Especially, as it was stated that in many low-income countries vaccinations are less likely to be adequately provided. Incorporating this into the study would have given it further depth and broadened the appeal of the work. 

Response: We appreciate the suggestion from the reviewer. We agree that a non-vaccinated parent that did not provide passive antibody to the chicks would better mimic the real-world scenario in low-income countries. These, however, were not available from our commercial source.  Additionally, this specific commercial hybrid layer would more typically be produced in mid-scale operations (not village flocks) in low-income countries, so would likely have vaccinated parents in that production system.

Specific comments

  1. Line 250-253: “Chickens infected with NDV show respiratory signs such as gasping and coughing; nervous signs such as tremors, paralyzed wings and legs, and twisted neck; swelling in the eyes and neck; watery diarrhea; and reduced egg and meat production”

This statement is missing references.

 Response: Three references have been added to the sentence. “Chickens infected with NDV show respiratory signs such as gasping and coughing; nervous signs such as muscle tremors, paralyzed wings and legs, and twisted neck; swelling in the eyes and neck [1]; watery diarrhea [2]; and reduced egg and meat production [3].” (Line 256-260)

  1. Line 265-266: “The immunosuppressive effect of the NDV vaccine has also manifested in downregulation of some immune pathways”

This statement assumes that the immunosuppressive effect is proven, if this is the case it should be previously stated. I would believe that a suggestive statement of “may cause an immunosuppressive effect” would be better suited unless there is also corresponding blood work provided.

Response: We appreciate the reviewer’s suggestion and have changed the sentence to “The downregulation of some immune pathways also suggests that the NDV may have caused an immunosuppressive effect.” (Line 276-277)   

  1. 293-294: “Inhibition of Rho family GTPases pathway has been shown to inhibit the cell-cell fusion induced by glycoprotein of respiratory syncytial virus”

Could this statement be expanded on? Perhaps what is inferred by cell-cell fusion, e.g wound healing or damage repair? And in addition, how this could be useful in this model?

Response: The statement has been expanded as following. “It has been reported that inactivation of RhoA signaling could inhibit the cell-cell fusion and syncytium formation induced by glycoprotein of respiratory syncytial virus which is a paramyxovirus [4]. Therefore, as a paramyxovirus, NDV may have its replication and infection compromised by downregulation of Rho family GTPases signaling. However, the inhibition of Rho family GTPases signaling may also impair the lymphocyte development, activation and migration [5], thus not beneficial for host immune defense.” (Line 304-309)

Reviewer 2 Report

this study provides better understanding for Newcastle disease virus induced immune response using unbiased assays for transcriptome analysis of spleen of infected chicks. I have some questions: 

1- I wish if its possible, the author shows the viral transcripts at the corresponding time points 2 and 6 DPI in spleen?

2. Regarding the RNA isolation and cDNA library preparation, did the author digest RNAs with DNase enzyme to get rid of the DNA contamination the samples?  

3. 526 DEGs were expressed at 2DPI and 36 at 6DPI, can you explain virus-induced downregulation of 490 DEGs at 6DPI?, how can you confirm that all DEGs are specific to lentogenic ND and not for other strains of ND or other viruses? i think other viruses or ND strains as control will be necessary ? 

4. in Table 2, can you add the control group as well, in terms of pathways?

5. Why did not the author combine several tissues to comprehensively charachterize host immune response?  

Author Response

1. I wish if its possible, the author shows the viral transcripts at the corresponding time points 2 and 6 DPI in spleen?

Response: We thank the reviewer for this great suggestion. We have now quantified transcripts from some NDV genes in the RNA sequences that were not mapped to the chicken genome.  Additions have been made to the Materials and Methods (Line 105-108), the Results (Line 145-146) and the Discussion (Line 260-266) sections.

2. Regarding the RNA isolation and cDNA library preparation, did the author digest RNAs with DNase enzyme to get rid of the DNA contamination the samples?  

Response: Yes. The isolated RNA samples were treated with DNase enzyme using TURBO DNA-free kit (Thermo Fisher Scientific) to get rid of the DNA contamination. This information has been added to the methods part in the manuscript (Line 85-86).

3. 526 DEGs were expressed at 2DPI and 36 at 6DPI, can you explain virus-induced downregulation of 490 DEGs at 6DPI?, how can you confirm that all DEGs are specific to lentogenic ND and not for other strains of ND or other viruses? i think other viruses or ND strains as control will be necessary ? 

Response: The DEGs are differentially expressed genes between NDV-challenged group vs. non challenged group at each time point. So the 526 DEGs at 2dpi do not completely overlap with the 36 DEGs at 6dpi, and the difference between the two time points are more than 490 genes. These genes do not show differential expression at 6dpi. This does not mean they are downregulated but indicates that the effect of NDV infection on their expression has become mild, in that there are hundreds fewer genes at 6 dpi that differ between the NDV-challenged and the non-challenged groups. Because this experiment is strictly controlled in animal biosafety level 2 room, there is no possibility for the chickens to get viruses other than the administered virus. We do not claim that the identified genes can only change in response to lentogenic NDV, because we did not in this study compare the response to other strains of NDV or to other viruses. It is possible that other NDV strains and other viruses induce changes in some of the same genes identified in the current study. The current study was designed to focus on lentogenic NDV-challenged versus the non-NDV-challenged states.

4. in Table 2, can you add the control group as well, in terms of pathways?

Response: The pathways in table 2 are those predicted to be significantly changed in the NDV-challenged group compared to the control, non-challenged group. So the pathways would be exactly the same for control group vs. challenged group, but just with opposite direction of change.

5. Why did not the author combine several tissues to comprehensively charachterize host immune response?  

Response: Funding limitations only allowed assessment of one tissue. We agree that analysis combining several tissues would generate a more comprehensive characterization of the host transcription response, and hope that is possible to conduct an expanded study in the future. This could be very informative because different tissues usually have different functions and responses to the virus.

Round 2

Reviewer 2 Report

Thanks for the contributing authors for taking considerations the previous comments.

Round 3

Reviewer 2 Report

i would like to thank the author for the rebuttal letter. They nicely cover the questions and edited the manuscript as required.